# OpenReview forum: "From Grunts to Lexicons: Emergent Language from Cooperative Foraging"
_ICLR.cc/2026/Conference — Submitted to ICLR 2026_

### Official Review · Reviewer_k44V · 2025-10-22

**Soundness:** 4
**Presentation:** 4
**Contribution:** 3
**Rating:** 6
**Confidence:** 5

**Summary:**

This paper introduces a gridworld foraging environment as a testbed for emergent communication. The environment supports two tasks: one where a pair of agents communicates to retrieve the highest valued object, and one where they communicate to retrieve objects in their spawning order. The communication channel, over a vocabulary of 4 symbols, is separate from the action space. The authors find that communication indeed emerges between agents and exhibits many properties present in human language: compositionality, spatial or temporal markers (example of displacement), interchangeability, and cultural transmission.

**Strengths:**

I enjoyed reading this paper. I found it to contextualize well within the emergent communication literature, and to go above and beyond experimentally, exploring the interaction of many variables (population size, embodiment, bidirectional channel, compositionality, etc) that are usually explored separately.

To me, **the highlight of the paper was the result on displacement**, which is nontrivial, and to the best of my knowledge, has not been shown before. I would definitely emphasize this finding (nothing wrong with the other findings, but they mostly confirm existing results). An interesting aspect of the spatial/temporal markers is that they are tied to **communicative need**, i.e., the fact that in Game II (where displacement emerges) time information was necessary to succeed at the game, whereas in Game I it was not. This ties well into theories that what gets lexicalized in language are things that are important to communicate about (cf. Gualdoni et al., 2024, Bickerton 2009).

Overall, I think the findings + environment are relevant to the emergent communication community, and I would recommend acceptance.

**Weaknesses:**

There were two main weaknesses, both of which affected my score. If both are addressed I will raise my score to an 8.

1. **Contextualizing results in literature**: Many of the results (except displacement) have been independently attested in the literature. For instance,
	1. Population size affects compositionality: shown in Rita et al., 2022.
	2. Implicit communication can emerge when explicit communication is disabled: shown in Mordatch and Abbeel
   I'm missing a discussion of how your results compare to these existing ones. In many cases "This corroborates results in CITATION, who did X and found Y" is probably sufficient.

2. **In the theoretical appendix:** Assumption 1 is unrealistic due to agent capacity being relatively unconstrained (see Tucker et al., 2022 for an example where capacity is constrained). We can construct a situation where N agents are equally multilingual in N languages, such that the language distance is high but performance is not sacrificed. Since the theoretical results depend on Assumption 1, I strongly recommend to remove the theoretical portion entirely and rework it for a future standalone submission.

**Questions:**

l088 add Lowe et al., 2020

l141 add [Bullard et al., 2020](https://arxiv.org/abs/2010.15896)

l224 "To encourage temporal and spatial displacement... " here, it would be helpful to elaborate on how this promotes displacement.

**Missing citations:**

Emergent communication for understanding human language evolution: What's missing? Galke et al, 2022

---

> ### Author Response · Authors · 2025-11-21
>
> We thank Reviewer k44V for the thoughtful, thorough, and encouraging review. We greatly appreciate your positive assessment of the paper’s scope, experimental depth, and especially your recognition of the displacement result. Your comments helped us clarify our relation to prior work, refine the positioning of our contributions, and strengthen the manuscript overall. We have revised the paper accordingly, addressed the missing citations, and added explanations where requested. All changes appear in **blue** in the updated manuscript. Detailed responses to each point follow below.
>
> ## k44V-Weakness-1 Contextualizing results in literature:
>
> We agree that we should additionally contextualize our results. We provide additional context in these paragraphs in the results section.
>
> For **Increasing population size and self-play training support interchangeable language**, we add the previous result in ```line 363```:
>
> In agreement with prior research, (Dubova & Moskvichev, 2020) found that both population training with $N_{pop} \geq 3$ and the use of SP facilitate the development of a common language.
>
> For **Implicit communication can emerge when explicit message communication is disabled**, we add the previous result in ```line 449```:
>
> The result is consistent with prior research. Mordatch & Abbeel (2018) found that agents can use their physical body to send signals to others (e.g., through actions like pushing or pointing).
>
> For **Population size affects compositionality**, we add the previous result in ```line 455 and 462```:
>
> While **topsim** trends align with and extend previous studies reporting decreases (Rita et al., 2022) or stability (Michel et al., 2023), the updated **repcom** results suggest that larger populations do not necessarily enhance representational compositionality, offering a complementary perspective on how community size shapes emergent communication.
>
> ## k44V-Weakness-2 Assumption 1 is unrealistic
> We agree that the theoretical analysis cannot explain the case where an agent learns multiple languages. We appreciate thoughtful feedback and consider removing it from the final manuscript.
>
> ## k44V-Suggestion-1 Emphasize displacement as core contribution
> We added this in core novel contributions in introduction (```line 122```)
> ## k44V-Suggestion-2-and-3 l088 add Lowe et al., 2020, l141 add Bullard et al., 2020
> Thank you for suggesting these amazing works. We added these two works in ```line 103``` and ```line 150``` in the updated version.
>
> ## k44V-Suggestion-4 l224 "To encourage temporal and spatial displacement... " here, it would be helpful to elaborate on how this promotes displacement.
>
> We elaborated on how it promotes displacement in ```line 237``` in the updated version. We are happy to make adjustments if the reviewer feels it is still unclear.
>
> ## k44V-Suggestion-5 Missing citations:  Emergent communication for understanding human language evolution: What's missing? Galke et al, 2022
> This work was cited in the original manuscript (```line 045``` in the updated version).

---

> ### Author Response · Authors · 2025-11-26
>
> Thank you again for a thoughtful review. As the discussion period ends next week, we wanted to gently ask if these updates address all concerns. We are happy to make further adjustments.

---

### Official Review · Reviewer_x9TL · 2025-10-27

**Soundness:** 2
**Presentation:** 3
**Contribution:** 2
**Rating:** 4
**Confidence:** 4

**Summary:**

The paper proposes to study emergent communication in a foraging environment. LSTM-based agents are placed into a 2D grid world where they are incentivized to cooperate and forage. The subject of interest is the communication protocol that emerges between the agents. The authors quantify the degree of compositional structure, to what extent agents speak a similar language, and argue why the setting of foraging is more representative of the emergence of language in our species, compared to referential games.

**Strengths:**

The paper is well motivated from the viewpoint of using this experimental setup for studying human language evolution.

The paper is well situated within the related literature.

The proposed framework of studying emergent communication in foraging games is compelling and conceptually intuitive.

The results give hints that this framework would lead to desirable phenomena to be reflected (e.g., population size effects).

The paper comes with a rich appendix, studying important factors such as vocabulary size and generalization to unseen positions.

**Weaknesses:**

Contribution. Foraging games / 2d grid worlds have been extensively studied in multi-agent reinforcement learning (as noted in the submitted version). I am surprised that this would be (as claimed) new to the field of emergent communication (?) My rating is based on the assumption that foraging was not studied in emergent (symbolic) communication before.

Baselines. The paper proposes a new experimental framework to study emergent communication. In this setting, a suitable baseline would be the standard referential game in order to compare the two. However, in its current form the paper lacks such a comparison.

Metrics. It has become a habit in the field of emergent communication that every paper uses topsim + some new ad-hoc defined metrics (since topsim is flawed by design). This paper at least re-uses one other metric than topsim – but again introduces new metrics. I would have preferred to see results for previously proposed metrics, e.g., the ones from Conklin & Smith (ICLR 2023) or Elmoznino et al. (ICML 2025).

Model architectures. Only LSTM-based agents are considered. While somewhat standard in the emergent communication literature, the findings would be more interesting if different architectures were considered, especially transformers.

**Questions:**

I noticed the main setup is designed to be cooperative and a non-cooperative version is reported in Appendix. However, the description there only ablates informativeness of messages through perturbation. Can you elaborate why this would implement a non-cooperative version of the game? It would be interesting to study e.g., deception, if communication was still functional but the game design would reward non-cooperative behavior.

Why does topsim decrease with an increase of vocabulary size?

---

> ### Author Response · Authors · 2025-11-21
>
> We thank Reviewer x9TL for the thoughtful evaluation. The comments on compositionality metrics and model architectures were particularly valuable and helped identify areas for strengthening the manuscript. Below, we address these points in detail and outline the corresponding revisions, which are highlighted in **blue** in the updated manuscript.
>
> ## x9TL-Weakness-1 Contribution of the environment: Foraging games
> Thank you for the comment. We do not claim to be the first to study emergent communication (EC) in embodied settings. In the revised introduction (```lines 051 and 072```), we now explicitly frame our work as a follow-up to Mordatch & Abbeel. Our contribution is to show that an embodied foraging environment provides a complementary alternative to standard referential games: while most recent EC studies still use referential setups, embodiment enables linguistic properties, especially displacement, that cannot arise otherwise.
>
> To avoid redundancy, please refer to our response to **MHxj Weakness-1** above, where we provide a detailed itemized comparison of the two settings.
>
>
> ## x9TL-Weakness-2 Referential Game Baselines
> We have a referential game version. The result is shown in Figure S8 and is mentioned in the main text (```line 416```). Briefly, when an agent cannot move and can remotely pick up an item, position information cannot be decoded from messages because it is not important for this task.
>
> ## x9TL-Weakness-3 Compositionality Metrics
> We added experiments using recently proposed **repcom** from Elmoznino et al. These results are now included in Figure 6a and Table 5S, and they provide new insights that were not visible with our original metric set. More broadly, we fully agree with your observation that relying on a single metric is insufficient.
>
>
> ## x9TL-Weakness-4 Model architectures
>
> While LSTM/GRU architectures are indeed standard in online RL and MARL settings, in part due to their stable, widely-tested implementations [26–28], we agree that exploring alternative ones can provide important insights into the robustness of emergent communication findings.
>
> Your comment motivated us to directly compare LSTM-based with Transformer-based agents (Sec E, ```line 1505```). This comparison strengthens the paper by showing that key trends, including interchangeability and cultural transmission, hold for both architectures, suggesting the findings are not tied to a specific model choice.
> ## x9TL-Question-1 Non-Cooperative Setting
> In the appendix, we approximate a non-cooperative setting by removing shared goals: each agent maximizes only its own reward, making communication strategically useless. Perturbation verifies that agents ignore messages under this setup. We agree that studying deception would be valuable, but it would require redesigning the game so that misleading a partner is actually rewarded—an interesting direction for future work.
> ## x9TL-Question-2 Why does topsim decrease with an increase of vocabulary size?
> Topsim decreases with increasing vocabulary size, possibly because a larger vocabulary weakens the information bottleneck that normally forces agents to build structured, reusable, compositional codes. With more available symbols, agents might solve the task using non-systematic message assignments. This may break the alignment between semantic distances and message distances, causing the topsim to fall.

---

> > ### Comment · Reviewer_x9TL · 2025-11-25
> >
> > I have read the author response, other reviews, and checked the updates in the paper. I greatly appreciate the contextualization with previous embodied EmeCom work, the inclusion of new measures (representational compositionality, positional disentanglement), and a variant of the experiment with the Transformer architecture. This strengthens the paper substantially and I will revise my scores accordingly.

---

> ### Author Response · Authors · 2025-11-26
>
> We thank Reviewer x9TL **for the positive re-evaluation to 8.** We are glad that the improved contextualisation, along with the Transformer-based architecture and the recently proposed representational compositionality metric, have substantially strengthened the work.

---

### Official Review · Reviewer_MHxj · 2025-10-30

**Soundness:** 1
**Presentation:** 1
**Contribution:** 2
**Rating:** 2
**Confidence:** 4

**Summary:**

This paper introduces Foraging Games (FG). FG is a multi-agent 5x5 gridworld for studying Emergent Communication. Agents can move in a partially-observed two dimensional grid, communicate discrete tokens over a communication channel (a part of the action space), and collect objects towards accomplishing an objective. There are two types of objectives in the game (each yielding a different environment):
- ScoreG: Two objects are given different scores, each agent knows one of the scores, the agents must simultaneously pick up the higher-scoring object. The agents cannot see each other.
- TemporalG: Objects are spawned over time in one of the agents' field of view. Agents must pick up the objects on an agreed-upon order.

The authors study the learning dynamics and the emergent communication protocol. In particular, experiments examine the following questions:
1. Does an effective communication protocol emerge?
2. Is the communication protocol compositional? This is done by measuring topographical similarity between messages and meanings.
3. The effect of population size. Here population refers to the number of instantiations of the architecture, e.g. population size $15$ means that at every round of training two of fifteen NN instantiations are taken as the agent policies.
4. The effect of the social network structure. For example, Self-Play means letting an agent policy play with itself (a loop in the network), a ring network means letting the $i$th player play with the $i + 1 mod N$ player, where $N$ is the population size. In addition, rings-with-cliques and a Watts-Strogatz networks are examined. In particular, the effect of the network structure on the ability of agents to generalize to new players (zero shot coordination).
5. What happens when the communication channel is turned off so that agents communicate purely through their actions.

**Strengths:**

- The investigation of network structure on properties of the emergent communication protocol are novel and very interesting in my opinion. I think this is the main contribution of the paper, and would suggest reframing the paper around it considering that many of the other ideas are not novel (see Weaknesses).
- The paper is well-situated in a growing body of literature on emergent communication. In particular, the game is similar to the setting of Mordatch and Abbeel (2017) which was a seminal work in the field.
- There is a rich and interesting collection of ablations.
- The various settings are generally well-presented. I was able to understand what the experimental setup and results easily.

**Weaknesses:**

- The Foraging Game setup is very similar to the setup explored by Mordatch and Abbeel (2017), in what is a foundational work in emergent communication. This was the first (or among the first) papers to study embodied emergent communication, specifically with agents moving on a (continuous) two dimensional world, without seeing each other, in order to reach landmarks (equiv. pick up objects) at a certain order. That paper explicitly studied compositionality. None of this is mentioned in the paper, which is the main reason for my presentation score (contextualization relative to prior work). Presenting this paper as a follow-up to Mordatch and Abbeel would be more honest with respect to originality; considering the impact of the 2017 paper, it would only strengthen this paper.
- The ablation on "Implicit Communication" is essentially changing the Emergent Communication setup to a Social Learning one (Ndousse et al., 2021 and several other papers around that time by Natasha Jaques). In MARL social learning, the question is whether agents learn from each other by observing each others' actions. The connection should be made explicit.
- The paper neglects to mention the rich body of literature on embodied emergent communication, indeed starting from Mordatch and Abbeel. In particular, the second paragraph in the introduction seems to imply that the study of embodied emergent communication is novel to this paper; that is far from the case.
- More broadly, the title, introduction and conclusion of the paper suggest that the paper has ramifications to the study of the evolution of language. The question of whether results emergent communication have meaning for natural language is a topic for heated debate [e.g. 1, 2, 3, 4]. In particular [1] argues that following Hockett's design feature (such as displacement, emphasized in this paper) do not necessarily bear significance for language evolution. This is the reason for my soundness score. I suggest instead that the authors reframe their paper as bearing meaning on training dynamics of MARL, rather than overpromising and underevaluating as a paper on natural language evolution.
- Minor: Citation styles are in the wrong format in most of the paper. Use citep instead of citet. The experiment shorthand notation (e.g. P3-FC-XP) is not helpful, because as a reader I am not familiar with the many acronyms.

[1] Language Evolution: Why Hockett’s Design Features are a Non-Starter. Wacewicz and Żywiczyński 2014.
[2] Natural Language Does Not Emerge ‘Naturally’ in Multi-Agent Dialog. Kottur, Mourra, Lee and Batra. 2017
[3] Measuring non-trivial compositionality in emergent communication. Korbak et al. 2020.
[4] Anti-efficient encoding in emergent communication. Chaabouni et al. 2021.

**Questions:**

- What is the "semantic space" used for measuring topographical similarity?
- Besides using a discrete state and action space, are there other significant ways in which Foraging Games differ from Mordatch and Abbeel's grounded communication environment?

---

> ### Author Response · Authors · 2025-11-21
>
> We thank Reviewer MHxj for a thoughtful review. Your comments helped us sharpen the framing and clarify the contributions of the paper. We have updated the manuscript accordingly, with all revisions marked in blue, and we respond to each point below.
>
> ## MHxj Weakness-1: Similarity to Mordatch & Abbeel (2018)
> We agree that their work is foundational in embodied emergent communication (EC). We now explicitly acknowledge their work as pioneering (```line 051```), summarizing its key elements: co-learning of physical and linguistic behavior, landmark-based coordination (```line 156```), and implicit signaling through movement (```line 448```).  We clarify the main differences in our setting below.
>
> **Novelty relative to Mordatch & Abbeel (2018)**
>
> **1. Displacement** TemporalG requires agents to communicate about past, unobservable events: one agent sees a spawn, must remember it, meet later, and convey time–location information, which their tasks do not target. (strength highlighted by ```Reviewer k44V and 7a3K```).
>
> **2. Partial observability:** Our agents see only a W×W local window, and key information is split across agents (e.g., item scores or spawn events). In Mordatch & Abbeel, agents observe the full environment, with uncertainty coming mainly from private goals rather than limited perception.
>
> **3. Multi-agent coordination:** Both settings require communication, but in Mordatch & Abbeel each agent can carry out its subgoal independently once informed. In our games, rewards depend on tightly coupled joint actions (e.g., simultaneous pickup), so agents must coordinate both information and action timing (solo completion is impossible).
>
> **4. Decentralized Learning**: they train a single shared policy via gradients through a differentiable environment. Our agents learn independently with separate networks and sparse rewards, without environment gradients, allowing heterogeneity.
>
> **5. Social Dynamics**:  We vary population size and social-network topology to study compositionality, language similarity, interchangeability, and cultural transmission. Their setup does not model population structure, self-interaction, or cultural dynamics.
>
> **6. Formal Compositionality:** We measure compositionality using **topsim** and the recently proposed **repcom** score, whereas they rely on vocabulary statistics and qualitative inspection rather than formal metrics.
>
> **7. Self-Play (SP):** We show that SP can promote cultural transmission in weakly-connected social networks, **not explored in prior work**.
>
> ## MHxj Weakness-2 Connection to Social Learning
> Our implicit communication follows the non-verbal signaling setup in Mordatch & Abbeel (2018). This could be related to, but **not the same as**, social learning in Ndousse et al. (2021). Social learning involves a novice observing a fixed expert, whereas in our setting, both agents must adapt their behaviour so that it becomes informative to a partner. The main difference is that a fixed expert does not adjust behaviour to be informative. Adding learnable signaling on the expert side would bridge the two formulations.
>
> ## MHxj Weakness-3 Missing Context on Prior Embodied EC Work
> See reply to **MHxj Weakness-1**.
>
> ## MHxj Weakness-4 Framing and Claims About Language Evolution
> Our framing **aligns well** with the work that the reviewer suggests, Wacewicz & Żywiczyński (2014), who argue that language-evolution research should move beyond code-level features toward modeling the cognitive mechanisms that generate them. At a computational level [25], we examine how memory, partial observability, shared intentions, and population structure can jointly produce displacement and cultural transmission, offering mechanistic hypotheses rather than a full model of human language.
>
> We also agree that EC should not be read as a direct account of how human language evolved. Our aim is to show how MARL settings can illuminate pressures that **may** shape language.
>
> While EC may differ from human language, it shares some statistical properties with human language, and recent work shows that EC-trained models can transfer to human language tasks [20–21]. The revised Discussion (```lines 481–484```) now emphasizes this **functional-analogue** interpretation, without overclaiming evolutionary significance. We also added a limitation (```line 506```), noting that Foraging Games lack the semantic breadth, syntax, and open-endedness. We are happy to make further adjustments if the reviewer feels a different framing would improve clarity.
>
> ## MHxj Weakness-5 Citation Style
> We fixed the style.
> ## MHxj Question-1 What is the semantic space?
> It is defined by the item state vectors $S = [x_i, y_i, s_i]$, where $\(x_i, y_i\)$ denote the location of item $\(i\)$, and $\(s_i\)$ denotes its score. We have provided more detail in the updated manuscript (```line 966```).
> ## MHxj Question-2 Environment different from Mordatch and Abbeel's?
> See **MHxj Weakness-1**.

---

> ### Author Response · Authors · 2025-11-26
>
> With the discussion closing soon, we wanted to check if our revision resolved your concerns regarding Mordatch & Abbeel (2018) and the framing of language evolution. We are happy to make further adjustments.

---

### Official Review · Reviewer_7a3K · 2025-10-31

**Soundness:** 3
**Presentation:** 2
**Contribution:** 1
**Rating:** 2
**Confidence:** 4

**Summary:**

This paper introduces the Foraging Games framework within a MARL setting to study the emergence of language under specific ecological constraints, including embodiment, partial observability, and the need for temporal reasoning. The authors successfully demonstrate that the resultant communication protocol exhibits key linguistic properties The work provides empirical evidence on how environment-driven constraints can lead to the emergence of specific language functions. While the study is technically sound and achieves its specific goals, the choice of research paradigm in the current AI landscape raises significant questions about its overall significance and generalizability.

**Strengths:**

1. The introduction of the Foraging Games (FG)  effectively integrates ecological constraints such as partial observability and the necessity of temporal reasoning. This moves beyond traditional, more static RefGame and offers a more ecologically valid setting for studying language origins.

2. The paper provides clear evidence for the emergence of both Interchangeability, arbitrariness, compositionality, cultrual transmission and, more notably, Displacement—the ability to refer to non-present facts (past events). This is a strong result supporting the hypothesis that temporal demands drive linguistic complexity.

**Weaknesses:**

Fundamental Critique on Research Significance and Paradigm:
This is the most critical weakness. The research paradigm employed—utilizing small LSTM models in a highly simplified, "toy" $5 \times 5$ environment—appears to be meaningless. In the age of LLMs which have demonstrated scalable compositionality and robust generalization, the conclusions drawn from this limited setting face major challenges in terms of relevance and transferability. This work provides little inspiration for improving algorithms used in modern LLMs or complex MARL systems. It is also inadequate for providing insights of evolutionary linguistics: The small population size ($N \le 15$) and limited training time are insufficient to simulate the long-term cultural transmission and evolutionary pressures necessary for drawing rigorous conclusions.

(1)  Limitations of Embodiment and Population ScaleWeak Embodiment: The $5 \times 5$ grid world is too simplistic to qualify as true embodiment. The task primarily constitutes a simplified partially observable planning problem, lacking the challenges of high-dimensional perception, complex motor control, and detailed physical interaction that define true embodied AI. This simplification limits the ability to observe how genuine physical constraints shape complex syntax.

(2) Insufficient Population Scale: The tiny population size ($N \le 15$) restricts the derived "language" to an arbitrary convention among a few agents, not a robust, social-driven language system. Conclusions regarding linguistic structure and robustness are fundamentally limited by the absence of the social pressure and generational learning required for large-scale conventionalization.

(3) Questionable Emergence of Displacement: The claim of emergent displacement is weakened by the possibility of explicit encoding, illustrated in Figure2.

(4) Limited Generalization and Structural Robustness Poor Generalization to Unseen Locations: The significant drop in success rate when generalizing to unseen object locations (e.g., 0.953 to 0.756 in ScoreG, as noted in the Appendix) suggests a failure to learn an abstract, coordinate-independent spatial reference grammar.

**Questions:**

Contemporary Significance: Given the current landscape of LLMs and large-scale MARL, what is the non-trivial research significance of studying EC in $5 \times 5$ grid worlds today? How do the specific architectural or algorithmic insights here transfer to models operating in high-dimensional, open-world settings?

Memory Structure and Displacement: Please clarify the exact architecture and input encoding of the agent's memory module ($M_t$). To what extent is temporal or spatial information explicitly encoded or tagged in the input observation ($O_t$) before it enters the RNN/GRU? This is crucial for verifying the genuine emergence of displacement versus a simple mapping/retrieval from a pre-structured memory.

---

> ### Author Response · Authors · 2025-11-21
>
> We thank Reviewer 7a3K for the thoughtful review and for recognizing our contribution to **cognitive-science research** on how ecological constraints shape core linguistic properties. Your comments have helped us clarify the scope, limitations, and significance of the work; we have revised the manuscript accordingly (changes in **blue**) and address each point below.
>
> ## 7a3K-General-Critique: Concerns About Research Significance and Toy Setting
> **Small models and simplified tasks:** Small models and simplified tasks: Using small models and controlled environments is a deliberate scientific choice. In emergent communication, MARL, and cognitive science, such “toy’’ settings are standard because they isolate causal mechanisms that large, entangled systems like LLMs obscure. Our goal is not to engineer better LLMs, but to understand how linguistic properties emerge under specific pressures. Simple models and foraging tasks provide a tractable platform for measuring displacement, compositionality, and cultural transmission without confounding factors.
>
> **Algorithmic contribution:** Our work does not aim to advance algorithms in modern LLMs or MARL systems. Instead, it seeks to uncover fundamental principles underlying the emergence of communication in MARL settings.
>
> **Toy environments still matter for LLMs:** Even state-of-the-art LLMs fail on simple theory-of-mind tests [1,2], rock–paper–scissors [3], and grid-world tasks [4–6]. These results show that controlled, interpretable environments remain essential for probing fundamental reasoning and communication. While our work does not target LLM evaluation, the value of toy environments can extend beyond small models.
>
> **Modelling Evolution of Language:**
> Our aim is not to model the full complexity of human language evolution, but to study key mechanisms highlighted in evolutionary-linguistics theories, especially shared intentions under partial observability. The Foraging Games instantiate these pressures in a controlled setting, letting us examine how linguistic properties can emerge from reward-driven multi-agent interaction without supervision or human data.
>
> ## 7a3K-Weakness-2: Limits of Embodiment due to Grid World
> We agree the environment is simplified, and we include larger grids in the appendix (Sec. D1). That said, simplicity has long been valuable in RL and MARL [7–9]: grid worlds like Overcooked, Minigrid, Level-Based Foraging, and Craftax are standard because they enable clear hypothesis testing and interpretability, while high-dimensional embodied settings often obscure mechanisms. Our aim is to study how linguistic structure emerges under controlled pressures, and grid worlds offer an appropriate abstraction, like the grid-like codes used in human brains [12].
>
>
> ## 7a3K-Weakness-3 Tiny population
>
> From an EC perspective, a population of N = 15 is large enough to reveal population-level effects [15]. From a linguistic perspective, small communities can still develop full compositional languages; Nicaraguan Sign Language is a classic example. From an MARL perspective, N = 15 is not small: fully decentralized training (FDT) already requires ~1B environment steps, and scaling further introduces severe non-stationarity, making FDT convergence unlikely [13,14].
> ## 7a3K-Weakness-4 Questionable Emergence of Displacement
> Our agents use a single-layer LSTM with no explicit temporal or spatial tags. Each timestep includes only the 3×3 local view, the agent’s grid position, and the partner’s message. The LSTM must infer temporal order and absolute locations from sequence dynamics, so displacement arises from learned memory use rather than built-in structure. When movement is removed (no-body variant), spatial information becomes non-decodable while score remains decodable (Fig. S8), showing that spatial encoding appears only when the task requires it.
>
> ## 7a3K-Weakness-5 Limited Generalization and Structural Robustness:
> Our new results show that the emergent language does encode unseen item states (Fig. S9). The performance drop instead reflects known MARL generalization limits [11]. Non-communicating agents with full observability show a similar decline on unseen layouts (Table S7), indicating that the gap stems from coordination strategy rather than a failure of the communication protocol.
> ## 7a3K-Question-1 Contemporary Significance of Emergent Communication
> Emergent communication (EC) continues to inform contemporary AI. In LLMs and VLMs, techniques first explored in EC, such as iterated learning, have been adapted to improve compositionality and strengthen out-of-distribution generalization [17]. In MARL, our environment exposes the limits of fully decentralized learning: even strong baselines like PPO converge slowly at larger population sizes. This makes our setting a useful benchmark for developing MARL methods that scale in coordination and communication.
>
> ## 7a3K-Question-2 Memory Structure and Displacement
> We refer to **7a3K-Weakness-4**.

---

> ### Author Response · Authors · 2025-11-26
>
> Thank you for recognizing the technical soundness of our work. As the discussion period ends next week, we would appreciate your feedback on whether these clarifications resolve your concerns regarding the research scope.
> We would like to gently remind you that the primary area of this submission is **Applications to Neuroscience & Cognitive Science**.

---

> > ### Comment · Reviewer_7a3K · 2025-11-27
> > **Reservations on the Significance of the Work**
> >
> > Thanks for the clarification. My confusion regarding Weakness-4, Weakness-5, and Question-2 has been resolved.
> >
> > My main point of concern is still regarding the significance of this work. I acknowledge the difficulty of scaling up the number of agents in MARL, the significance of small-scale grid worlds for studying communicative MARL, and now I am aware that the paper's track is "Applications to Neuroscience & Cognitive Science" and that EC has historically inspired AI while toy-scale settings are common, especially since I also did research in this field. However, I simply believe that when we are already seeing practical emergent phenomena in larger language models within larger scenarios, the significance of this type of research now appears diminished. The conclusions drawn on a small scale are too limited and lack practical significance.
> >
> > Regarding "the fundamental principles" about "how linguistic properties emerge under specific pressures," what insight can be gained if it is not derived from a certain model, task, or population scale? After all, a necessary condition for emergence is scale, or rather, while simple principles derived from a small scale may be valid, they can hardly be called new scientific discoveries. For instance, your claim that "Even state-of-the-art LLMs fail on simple theory-of-mind tests" is mostly based on conclusions from two years ago. To explore the limitations of LLMs on ToM, more complex experiments must be designed. Even if LLMs have limitations on certain toy cognitive problems, it does not justify the significance of your simple setup.
> >
> > I have raised my score specifically concerning the technical details. However, I''l keep it below the acceptance threshold regarding the significance of this work.

---

> ### Author Response · Authors · 2025-11-27
> **Training LLMs on massive data won't give any insight to evolutionary processes.**
>
> We thank the reviewer for the discussion and for **re-evaluating the score to 4**. We totally agree that increasing the complexity of the environment and scaling up the models would result in language with richer semantics. We have acknowledged this limitation in our revised manuscript (```line 506```). However, scaling up to truly open-ended tasks remains an open challenge in RL even without the added complexity of emergent communication. **We view our work as a necessary foundational step before such scaling is possible**. It is also important to note that **small grid-world environments are still a standard for evaluating MARL**, especially the studies that require many ablations (for example, OvercookedV2, ICLR'25).
>
> However, we have to **push back on the premise that the existence of large-scale LLMs diminishes the significance of this work** for the following reasons:
>
> - **It is difficult to claim that linguistic properties emerge in LLMs**. Since human language is already highly structured and compositional, explicitly training a model on massive amounts of it and then saying "look, structure emerged" could be misleading. The model is likely just mimicking the statistical patterns it saw in its training data, rather than developing a functional adaptation.
>
> - In contrast, **our agents start with nothing more than reward**. They do not have an answer key. Consequently, if they develop a way to combine tokens or talk about task-relevant things, we know for a fact that they invented it to solve a problem, not because they copied it from a dataset.
>
> - This allows us to prove why language develops specific features, which is a causal question that models trained on massive amounts of existing text cannot answer.
>
> - Finally, LLMs model the product of language evolution (the end result), whereas our work models the process (how we got there). **They serve the different but complementary goals.**
>
> Please let us know if this could resolve the significance issue.

---

> > ### Comment · Reviewer_7a3K · 2025-11-27
> >
> > Thanks for the reply. I agree that:
> >
> > Scaling up to truly open-ended tasks remains an open challenge for EC, without the help of existing corpus. LLMs largely benefit from the large training data, while EC 's problem setting is different.
> >
> > However, my main point of mentioning LLM is that: since we already see the potential of language emergence of the large models, the research of EC should also be on that way. Otherwise the contribution is limited.

---

> ### Author Response · Authors · 2025-11-27
> **This criterion would invalidate the entire field of foundational MARL research, not just our work**
>
> We appreciate the vision the reviewer suggests, but **we must ground expectations in the current state-of-the-art for RL and MARL**, as it is the core methodology of our work:
>
> - In RL, **scale is also defined by experience (time steps), not just model size**. Our agents interact for nearly **1 billion time steps** to evolve a shared, interchangeable language at $N_{pop}=15$ for ScoreG and $N_{pop}=3$ for TemporalG. This aligns with the "Serial Scaling Hypothesis" [32]: just as AlphaGo achieved "large-scale" results through massive experience rather than massive model size, our emergent language is a result of scaling interaction, not parameters.
>
> -  Our 15 agents have a joint message space of size $V^{15 \times L}$ (where $V$ is vocab size and $L$ is episode length). Achieving convergence to a shared language in this massive space without shared gradients is a significant challenge that relies on serially scaling interaction, not parameters.
>
> - Training even a "tiny" (in LLM sense) 1B-parameter model from scratch via RL requires industrial-scale compute (e.g., 256 TPUs as noted in [31]). MARL is significantly more computationally demanding due to non-stationarity; consequently, **there are no existing MARL models trained from scratch at this parameter scale**.
>
> - Current open-ended environments do not require or benefit from billion-parameter models.  State-of-the-art results on benchmarks like Craftax are achieved with models using only a few millions of parameters.
>
> - If the LLM-scale model is the new threshold for acceptance, **this criterion would invalidate the entire field of foundational MARL research, not just our work.**
>
> - Following the reviewer's suggestion, we added more complexity to the environment by procedural generation (```line 1653 in Sec. F```). The most complex environment can have **6 quadrillion unique layouts**. Moreover, we scale up the model to be 335 times bigger and compare its performance to the original one. First, none of the models achieve high performance when the environment reaches the maximum complexity. Second, our task does not benefit from parallel scale-up, aligned with the theory proposed in [32] and results reported in [33].

---

### Author Response · Authors · 2025-11-21

# Reference
[1] Large Language Models Fail on Trivial Alterations to Theory-of-Mind Tasks
[2] Clever Hans or Neural Theory of Mind? Stress Testing Social Reasoning in Large Language Models (EACL’24)
[3] Position: Theory of Mind Benchmarks are Broken for Large Language Models (ICML’25)
[4] Benchmarking World-Model Learning
[5] From Black Boxes to Transparent Minds: Evaluating and Enhancing the Theory of Mind in Multimodal Large Language Models (ICML’25)
[6] Towards A Holistic Landscape of Situated Theory of Mind in Large Language Models (ACL’23)
[7] JaxMARL: Multi-Agent RL Environments and Algorithms in JAX (NeurIPS’24)
[8] Craftax: A Lightning-Fast Benchmark for Open-Ended Reinforcement Learning (ICML’24)
[9] OvercookedV2: Rethinking Overcooked for Zero-Shot Coordination (ICLR’25)
[10] Shared experience actor-critic for multi-agent reinforcement learning (NeurIPS’20)
[11] The Overcooked Generalisation Challenge: Evaluating Cooperation with Novel Partners in Unknown Environments Using Unsupervised Environment Design (TMLR’25)
[12] The Tolman-Eichenbaum Machine: Unifying Space and Relational Memory through Generalization in the Hippocampal Formation (Cell’20)
[13] In-Context Fully Decentralized Cooperative Multi-Agent Reinforcement Learning (NeurIPS’25)
[14] Trust region bounds for decentralized ppo under non-stationarity (AAMAS’23)
[15] REVISITING POPULATIONS IN MULTI-AGENT COMMUNICATION (ICLR’23)
[16] On the role of population heterogeneity in emergent communication (ICLR’22)
[17] Ease-of-Teaching and Language Structure from Emergent Communication (NeurIPS’19)
[18] Iterated Learning Improves Compositionality in Large Vision-Language Models (CVPR’24)
[19] Language Evolution: Why Hockett’s Design Features are a Non-Starter. Wacewicz and Żywiczyński 2014.
[20] Searching for the Most Human-like Emergent Language (EMNLP’25)
[21] Linking Emergent and Natural Languages via Corpus Transfer (ICLR’22)
[22] Natural Language Does Not Emerge ‘Naturally’ in Multi-Agent Dialog (EMNLP’17)
[23] Measuring non-trivial compositionality in emergent communication. Korbak et al. 2020.
[24] Anti-efficient encoding in emergent communication (NeurIPS'19)
[25] From Understanding Computation to Understanding Neural Circuitry
[26] Stable-baselines3: Reliable reinforcement learning implementations (JMLR’21)
[27] SKRL: Modular and Flexible Library for Reinforcement Learning (JMLR’23)
[28] CleanRL: High-quality Single-file Implementations of Deep Reinforcement Learning Algorithms (JMLR’22)
[29] Towards a Formal Theory of Representational Compositionality (ICML’25)
[30] Compositionality with Variation Reliably Emerges in Neural Networks (ICLR’23)
[31] A Generalist Agent (TMLR)
[32] The Serial Scaling Hypothesis (Under review ICLR'26)
[33] 1000 Layer Networks for Self-Supervised RL: Scaling Depth Can Enable New Goal-Reaching Capabilities (NeurIPS'25)

---

### Meta-Review · Area_Chair_qCNR · 2026-01-07

**Summary:**

reviewers gave 2,2,4,6, with several highlighting key concerns for this paper:

Mostly toy environments and models, which limits the research impact.

Insufficient population scale to draw meaningful conclusions on emergent language.

Limited generalization and robustness to new locations and environments.

Concerns of novelty compared to other works in MARL, emergent communication in embodied settings, social learning and MARL, etc.

Lack of baselines in the newly proposed experimental framework to study emergent communication.

Lack of metrics from established works.

**Reviewer Concerns:**

Mostly toy environments and models, which limits the research impact.

--> gave reasons that this was by design. unclear how to resolve this concerns. reviewers did not seem to be satisfied in the end, seems outstanding.

Insufficient population scale to draw meaningful conclusions on emergent language.

--> same as above, gave reasons that this was by design. unclear how to resolve this concerns. reviewers did not seem to be satisfied in the end, seems outstanding.

Limited generalization and robustness to new locations and environments.

--> seems addressed

Concerns of novelty compared to other works in MARL, emergent communication in embodied settings, social learning and MARL, etc.

--> gave some explanations and reworked some parts of the intro and related work, but this to me is a serious concern, since not being honest and fair to related work in the first place calls into validity the entire work. concern seems outstanding.

Lack of baselines in the newly proposed experimental framework to study emergent communication.

--> authors explained that there is 1 ablation of their approach, which is not a baseline in my opinion (ie comparing to other work or modifications of other work). concern seems outstanding.

Lack of metrics from established works.

--> added this repcom metric and new results, seems to be resolved.

**Reviewer Scores:**

7a3K and MHxj might increase 2 to 4, but unlikely to lean positive.

x9TL might go from 4 to 6.

overall ratings will still be negative/borderline

---

### Decision · Program_Chairs · 2026-01-26

Reject